# Analysis of Newcastle disease virus prevalence in wild birds reveals interhost transmission of genotype VI strains

Tingting Zeng,[1,2] Liji Xie,[1,2] Zhixun Xie,[1,2] Jun Hua,[1,2] Jiaoling Huang,[1,2] Zhiqin Xie,[1,2] Yanfang Zhang,[1,2] Minxiu Zhang,[1,2,3] Sisi Luo,[1,2,3] Meng Li,[1,2] Can Wang,[1,2] Lijun Wan,[1,2] Houxun Ya[1,2]

**ABSTRACT** Newcastle disease virus (NDV) can infect more than 200 species of birds, and most reports have focused on poultry. The pathogenic ecology of NDV in wild birds remains largely unknown. Thus, an investigation of NDV in domestic wild birds and pigeons was conducted. Thirty-nine NDV strains were characterized from 3,549 oropharyngeal and cloacal swab samples collected from 2016 to 2019 in Guangxi, China. Among them, 3 NDV strains belonged to class I genotype 1.1.2, and 36 strains belonged to class II, including 6 strains belonging to genotype I, 1 strain belonging to genotype II, 4 strains belonging to genotype VI.2.1.1.2.1, 23 strains belonging to genotype VI.2.1.1.2.2, and 2 strains belonging to genotype XII. Phylogenetic analysis revealed that genotype VI NDVs circulate among multiple species of resident wild birds and may spread back to pigeons. In addition, resident wild birds are important for short-distance NDVs transmission. The prevalence of NDVs in wild birds indicates that more restrictive biosecurity measures and ongoing epidemiological investigations are needed.

**IMPORTANCE** Surveillance of Newcastle disease viruses (NDVs) has been conducted primarily in poultry, but their prevalence in wild birds remains largely unknown. Increasing our understanding of the ecology and evolution of NDVs in different species of birds will help us develop better prevention and control strategies. In this study, large-scale epidemiological sampling of resident wild birds in Guangxi from 2016 to 2019 was performed. The results demonstrated that different genotypes, especially genotype VI NDVs, circulated among resident wild birds. Our findings highlight the potential threat to the pigeon industry and public health.

**KEYWORDS** Newcastle disease virus (NDV), genotype VI, wild birds, interhost transmission

Newcastle disease virus (NDV) class II genotype VI strains normally cause infection and clinical disease in species of the Columbidae family, and this virus is often referred to as pigeon paramyxovirus 1 (PPMV-1) (1). At present, the population of breeding pigeons in China is approximately 40 million pairs, and approximately 700 million pigeons are sold for meat every year. PPMV-1 infection in pigeon flocks can lead to considerable economic loss owing to morbidity and mortality among birds and the prevalence of this virus worldwide (2–4). In a previous study, the prevalence of PPMV-1 in 12 provinces from 2014 to 2021 in China ranged from 0.50% to 3.19% in live bird markets (LBMs) (5).

Although 236 species of birds were reported to be naturally infected by NDV (6), nearly all genotype VI NDV strains were found in Columbiformes after host-switching events (7). In recent years, many wild bird-origin PPMV-1 strains have been reported in different countries and different birds of the Columbidae family, mainly rock pigeons, turtle doves, and Eurasian collared doves (8–10). The prevalence of NDV in other wild

**Peer Reviewer** Hassanein Hassan Abozeid, Cairo University, Giza, Egypt

Address correspondence to Liji Xie, xie3120371@163.com, or Zhixun Xie, xiezhixun@126.com.

The authors declare no conflict of interest.

See the funding table on p. 5.

birds remains largely unknown. A large-scale investigation of NDVs among domestic wild birds and pigeons was conducted in Guangxi from 2016–2019. Pathogenic tests to evaluate the virulence of the purified strains and phylogenetic analysis were performed to improve our understanding of the ecology and evolution of NDVs in different species of birds.

First, an investigation of NDVs was conducted, for which 3,549 oropharyngeal and cloacal swab samples were collected from domestic wild birds and pigeons in LBMs, and 20 organ samples were collected from diseased pigeon flocks in Guangxi from 2016 to 2019; these samples included 13 species and 9 orders (the sample and isolation details are provided in Table S1). The samples were suspended in phosphate-buffered saline (PBS) supplemented with antibiotics and then injected into 9-day-old specific pathogen-free (SPF) embryonated chicken eggs (Beijing Merial Vital Laboratory Animal Technology Co., Ltd., China) to isolate the viruses (11). After hemagglutination (HA)–inhibition (HI) tests of the harvested allantoic fluids, 39 NDV strains were obtained (12).

The full-length *F* gene was amplified via previously described primers and subsequently cloned and inserted into the pMD-18T vector for sequencing (13). To characterize these strains, a maximum likelihood (ML) tree was generated by using full-length *F* gene sequences with a pilot tree (14). According to the trees, 3 NDV strains belonged to class I (one from a pigeon collected from an LBM and two from francolins) (Table S2; Fig. 1A); 36 strains belonged to class II, including 6 strains belonging to genotype I (three from feral pigeons, two from francolins, and one from a turtle dove); 1 strain belonged to genotype II; 4 strains belonged to genotype VI.2.1.1.2.1 (two from diseased feral pigeons, one from a feral pigeon in an LBM, and one from a quail); 23 strains belonged to genotype VI.2.1.1.2.2 (four from feral pigeons in an LBM, one from a diseased feral pigeon, nine from turtle doves, five from spotted doves, two from pheasants, and two from quails); and 2 strains belonged to genotype XII (both from francolins). In summary, 27 strains belonged to genotype VI, representing 68% of the total isolates (Table S2; Fig. 1B).

Four passages for plaque purification in DF-1 cells were conducted to obtain pure genotype VI NDV strains (15). Determination of the mean death time (MDT) in 9‐day‐old SPF chicken embryos and an intracerebral pathogenicity test (ICPI) in 1‐day‐old SPF chicks following the World Organization for Animal Health (OIE) protocol were conducted to evaluate the virulence of the purified strains (11). Among the 27 strains, according to the OIE criterion, "ICPI ≥0.7 is considered virulent", 20 strains were virulent strains, and 7 strains had low virulence (Table S2). The cleavage sites in the F protein were all 112RRQKR↓F117, which is a molecular characteristic of virulent strains.

The whole-genome sequences of 27 strains were determined via next-generation sequencing on the Illumina HiSeq 2500 platform with $2 \times 150$-bp paired-end sequencing with an average depth of 200× (Zeta Biosciences, Shanghai, China). To better understand the evolution and host dynamics of these genotype VI strains, 94 full-length *F* genes, including those of 27 strains as well as those of other genotype VI.2.1.1.2.1 and VI.2.1.1.2.2 reference strains from South China since 2010, were used to construct a maximum clade credibility (MCC) phylogenetic tree, and transmission among the hosts was analyzed by Bayesian stochastic search variable selection (BSSVS) analysis via BEAST V1.10.4 (16). An uncorrected log-normal relaxed molecular clock model, a logistic growth tree model with GTR + G + I substitution, and a site heterogeneity model were used. A 300 million-state Markov chain Monte Carlo (MCMC) run was used, and sampling was performed every 10,000 states, with a 10% burn-in, to generate the MCC tree, which produced an effective sample size (ESS) of greater than 200 for every parameter.

According to the phylogenetic analysis, 4 genotype VI.2.1.1.2.1 strains formed a clade, and notably, 21 genotype VI.2.1.1.2.2 strains formed a clade (the clade is framed with a black dotted line Fig. 2A), with 3 strains from pigeons in Guangdong, Yunnan Provinces and Guangxi, and 2 strains from wild birds in Guangdong Province (17). The most likely ancestral host of this clade was the pigeon, and the most likely intermediate host was the turtledove; then, the viruses spread to seven host species (colors of branches indicate the

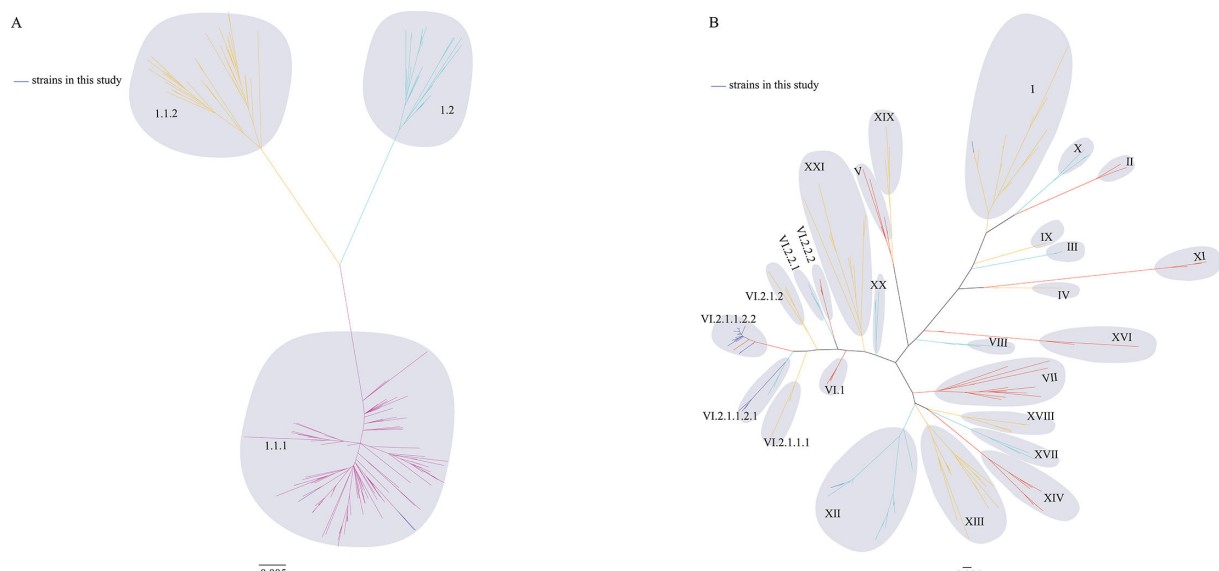

**FIG 1** Genotypes of the strains identified in this study characterized by an ML tree. (A) ML tree of the class I NDV strains. The purple line indicates the strains isolated in this study. (B) ML tree of the class II NDV strains. The purple line indicates the strains isolated in this study.

most likely ancestral host in Fig. 2A). Further BSSVS analysis revealed that the transition from pigeon to turtledove had very strong support from the Bayes factor (BF) (100 < BF < 1,000); the transitions from pigeon to magpie and from turtledove to pigeon and spotted dove had strong support (10 < BF < 100); and the transitions from pigeon to quail, turtle dove to quail, European turtle dove and grayheron, spotted dove to quail and pheasant, grayheron to European turtle dove, quail to pheasant, and European turtle dove to grayheron were also supported (3 < BF < 10) (18) (Fig. 2B). These phylogenetic analyses indicated that genotype VI.2.1.1.2.1 and VI.2.1.1.2.2 NDV strains may spill over from pigeon flocks to wild birds and circulate among wild birds, possibly even spreading back to pigeons.

The long-distance transmission of PPMV-1 was more likely related to pigeon transportation, commercial trade, or showing (7, 19). In this study, the wild bird source clade also included strains from Guangdong, Yunnan Provinces and Guangxi, which are geographically connected. Therefore, similar to our previous study, the finding suggested that resident wild birds play an important role in the short-distance transmission of NDV (12). All the wild birds sampled in this study appeared symptomless, indicating that the pathogenicity of PPMV-1 was weaker in wild birds than in pigeons, which also helped them transmit the virus in a proximal region.

In contrast to the results of other studies in which genotype VI NDV strains were isolated from migratory wild birds (17), the NDV strains in this study were all isolated from domestic pigeons and resident wild birds. These resident wild birds usually live around poultry industry facilities. The diverse dynamics of genotype VI NDV strains among resident wild birds may provide a greater chance to spread the virus back to pigeons, which are the species that are most susceptible to genotype VI NDV. Considering the weaker biosecurity of pigeon industry facilities than that of chicken or swine facilities, this may be one reason for the frequent occurrence of genotype VI infection in domestic pigeon flocks.

On the other hand, PPMV-1 reportedly caused a fatal respiratory disease in a human patient receiving immunosuppressive therapy (20). PPMV-1 may pose a public health risk because of its circulation among pigeons and resident wild birds that live around human communities. More restrictive biosecurity measures, such as the use of antibird nets in pigeon houses, are needed to prevent virus spillover from pigeons into the environment, and ongoing epidemiological investigations are needed.

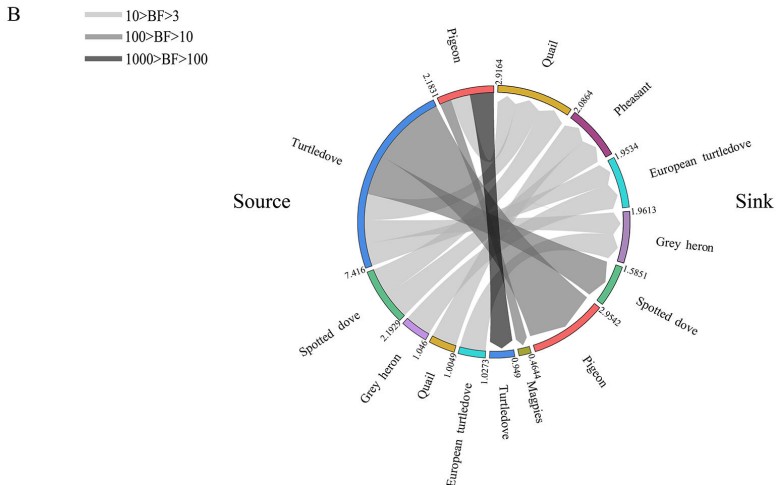

**FIG 2** Evolution and host dynamics of 29 genotype VI strains. (A) Maximum clade credibility phylogenetic trees of 94 genotype VI NDVs from South China collected since 2010. Subgenotypes are indicated with vertical gray bars, the strains isolated in this study are indicated with dark blue stars, and different hosts are indicated with different colors. The most likely ancestral hosts are indicated with colored branches. The time scale is shown at the bottom. (B) Transition rates

**Fig 2 (Continued)**

among hosts. The transition rates were generated from BSSVS analysis. The left side indicates the source, and the right side indicates the sink. Arrows in different shades of gray indicate the Bayes factor (BF) sections. The numbers outside the circle indicate the transition rates.

In conclusion, different genotypes, primarily genotype VI NDV, are circulating among multiple species of resident wild birds and may spread back to pigeons. Resident wild birds play an important role in short-distance NDV transmission. More restrictive biosecurity measures and ongoing epidemiological investigations are needed.

## ACKNOWLEDGMENTS

This research project was funded by the Natural Science Foundation of Guangxi (2024GXNSFBA010362), Guangxi Science and Technology Projects (AB16380054), Guangxi BaGui Scholars Program Foundation (2019A50), and Guangxi Shi Bai Qian Talents Engineering Foundation ([2020]24).

## AUTHOR AFFILIATIONS

[1]Guangxi Key Laboratory of Veterinary Biotechnology, Guangxi Veterinary Research Institute, Nanning, Guangxi, China
[2]Key Laboratory of China (Guangxi)-ASEAN Cross-border Animal Disease Prevention and Control, Ministry of Agriculture and Rural Affairs of China, Nanning, Guangxi, China
[3]College of Animal Science and Technology, Guangxi University, Nanning, Guangxi, China

## AUTHOR ORCIDs

Tingting Zeng http://orcid.org/0000-0002-6038-2843
Liji Xie http://orcid.org/0000-0003-0295-832X
Zhixun Xie http://orcid.org/0000-0002-1924-9952

## FUNDING

| Funder | Grant(s) | Author(s) |
| --- | --- | --- |
| 广西壮族自治区科学技术厅 \| Natural Science Foundation of Guangxi Zhuang Autonomous Region (Guangxi Natural Science Foundation) | 2024GXNSFBA010362 | Tingting Zeng |
| Scientific Research and Technology Development Program of Guangxi Zhuang Autonomous Region (Guangxi Science and Technology Planning Project) | AB16380054 | Liji Xie |
| Bagui Scholars Program of Guangxi Zhuang Autonomous Region (Bagui Scholars Program of Guangxi) | 2019A50 | Zhixun Xie |
| Guangxi Shi Bai Qian Talents Engineering Foundation | [2020]24 | Liji Xie |

## DATA AVAILABILITY

The whole-genome sequences of 29 NDV strains (class II genotype VI and genotype XII) and full-length *F* gene sequences of 10 NDV strains (class I and class II genotype I) were submitted to the NCBI. The accession numbers are listed in Table S2. The names and accession numbers of the reference strains are listed in Tables S3-S5.

## ADDITIONAL FILES

The following material is available online.

## Supplemental Material

**Supplemental material (Spectrum00816-24-S0001.docx).** Tables S1 to S5.

## Open Peer Review

**PEER REVIEW HISTORY (review-history.pdf).** An accounting of the reviewer comments and feedback.

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
