## [Reviewer comments · Microbiology Spectrum]

Microbiology Spectrum

Analysis of Newcastle disease virus prevalence in wild birds reveals interhost transmission of genotype VI strains

Tingting Zeng, Liji Xie, Zhixun Xie, Jun Hua, Jiaoling Huang, Zhiqin Xie, Yanfang Zhang, Minxiu Zhang, sisi Luo, Meng Li, Can Wang, Lijun Wan, and Houxun Ya

Corresponding Author(s): Liji Xie, Guangxi Veterinary Research Institute

Review Timeline:

Submission Date:	May 28, 2024
Editorial Decision:	July 29, 2024
Revision Received:	August 7, 2024
Accepted:	August 21, 2024

Editor: Dhammika Navarathna

Reviewer(s): Disclosure of reviewer identity is with reference to reviewer comments included in decision letter(s). The following individuals involved in review of your submission have agreed to reveal their identity: Hassanein Hassan Abozeid (Reviewer #2)

Transaction Report:

DOI: <https://doi.org/10.1128/spectrum.00816-24>

Re: Spectrum00816-24 (Analysis of Newcastle disease virus prevalence in wild birds reveals interhost transmission of genotype VI strains)

Dear Dr. Liji Xie:

Thank you for the privilege of reviewing your work. Below you will find my comments, instructions from the Spectrum editorial office, and the reviewer comments.

Revision Guidelines

Sincerely,
Dhammika Navarathna
Editor
Microbiology Spectrum

Reviewer #1 (Comments for the Author):

This manuscript reports investigation of NDV prevalence in wild birds and identification of interhost transmission of genotype IV strains. The authors tested 3549 oropharyngeal and cloacal swab samples for NDV and 39 were isolated and characterized in the present study.

The Abstract needs extensively rewriting. Lines 21-22 need writing how many were other genotypes in class II and how many were class I.

Lines 67-68, how HI is used to determine virus isolation as NDV, please cite related reference.

Lines 69-70, how the F gene was sequenced, please describe it.

Lines 90-91, which platform of NGS was used, please provide detailed information.

The whole manuscript need extensive check for language and grammar issues.

Reviewer #2 (Comments for the Author):

Zeng et al, have isolated and characterized 39 NDV strains from domestic wild birds and pigeons in Guangxi Province during 2016-2019, with 27 isolates belonging to genotype VI. The authors further characterized the pathogenicity of these isolates. In addition, they analyzed their transmission among the hosts based on the full-length F gene, revealing interhost transmission. The manuscript is well-written. The methodology is sound, and conclusions are justified. I have minor comments as follow:

- Abstract: Please mention the breakdown classification of the 39 NDV strains.
- Please mention the references for the viral isolation, HI, Plaque purification. Also please include the reference for the OIE protocol followed for the MDT and ICPI in the reference list.
- Lines 87-89: Why the virulence was between mesogenic and lentogenic while the F protein cleavage site was characteristic of virulent strains? The table shows most of the isolates as velogenic (ICPI >0.7). I think the authors confused lentogenic with velogenic.
- Line 134: "Considering the weaker biosecurity in pigeon industry". Weaker than what?
- Please add a conclusion statement at the end.

Executive Editor,

Microbiology Spectrum

Subject: Revised manuscript

Dear Editors,

Thank you for providing us with the reviewers' and your thoughtful comments on our manuscript entitled "Analysis of Newcastle disease virus prevalence in wild birds reveals interhost transmission of genotype VI strains". We are appreciative of the reviewers for thoroughly reviewing our manuscript. We feel that our manuscript has been greatly improved by the inclusion of their suggested changes. According to the reviewers' comments, we have revised the manuscript extensively. Here, we provide point-by-point responses to the reviewers' comments and highlight the changes that we have made. All of the modifications are marked in red in the revised manuscript.

Thank you very much for your time and consideration.

Sincerely,

Dr. Liji Xie, Professor

Guangxi Key Laboratory of Animal Vaccines and Diagnostics

Guangxi Veterinary Research Institute

51 You Ai North Road, Nanning, 530001, China

Tel: +867713120371

Fax: +867713120371

E-mail address: xie3120371@126.com

Responses to the reviewer's comments:

Thank you for your review. We tried our best to improve the manuscript and made some changes to the manuscript.

Reviewer #1 (Comments for the Author):

This manuscript reports investigation of NDV prevalence in wild birds and identification of interhost transmission of genotype IV strains. The authors tested 3549 oropharyngeal and cloacal swab samples for NDV and 39 were isolated and characterized in the present study.

The Abstract needs extensively rewriting. Lines 21-22 need writing how many were other genotypes in class II and how many were class I.

Response: Thank you for the suggestion to make our description clearer. Descriptions of the numbers of genotypes in class I and class II have been added. "Among them, 3 NDV strains belonged to class I genotype 1.1.2, and 36 strains belonged to class II, including 6 strains belonging to genotype I, 1 strain belonging to genotype II, 4 strains belonging to genotype VI.2.1.1.2.1, 23 strains belonging to genotype VI.2.1.1.2.2 and 2 strains belonging to genotype XII."

Lines 67-68, how HI is used to determine virus isolation as NDV, please cite related reference.

Response: Thank you for your guidance, we have updated and added relevant references as your suggestion. A reference describing how HI is used to identify an isolated virus as NDV has been cited.

Lines 69-70, how the F gene was sequenced, please describe it.

Response: Thank you for your comment. We have carefully made the improvements to address the issue. The following sentence has been added to briefly describe the F gene sequencing method, and the relevant reference has been cited: "The full-length *F* gene was amplified via previously described primers and subsequently cloned and inserted into the pMD-18T vector for sequencing".

Lines 90-91, which platform of NGS was used, please provide detailed information.

Response: Thank you for your comment. We have carefully made the improvements to address the issue. A sentence providing the details of NGS sequencing, including the NGS platform, has been added: "The whole-genome sequences of 27 strains were determined via next-generation sequencing on the Illumina HiSeq 2500 platform with

2×150 bp paired-end sequencing with an average depth of 200× (Zeta Biosciences, Shanghai, China).”

The whole manuscript need extensive check for language and grammar issues.

Response: Thank you for the suggestion. We have tried our best to polish the language in the revised manuscript and let the AJE language editing company to help us to improve our manuscript.

Thank you for your valuable feedback. We have carefully reviewed your comment and revised the manuscript.

Reviewer #2 (Comments for the Author):

Zeng et al, have isolated and characterized 39 NDV strains from domestic wild birds and pigeons in Guangxi Province during 2016-2019, with 27 isolates belonging to genotype VI. The authors further characterized the pathogenicity of these isolates. In addition, they analyzed their transmission among the hosts based on the full-length F gene, revealing interhost transmission. The manuscript is well-written. The methodology is sound, and conclusions are justified. I have minor comments as follow:

- Abstract: Please mention the breakdown classification of the 39 NDV strains.

Response: Thank you for the suggestion to make our description clearer. Descriptions of the numbers of genotypes in class I and class II have been added. “Among them, 3 NDV strains belonged to class I genotype 1.1.2, and 36 strains belonged to class II, including 6 strains belonging to genotype I, 1 strain belonging to genotype II, 4 strains belonging to genotype VI.2.1.1.2.1, 23 strains belonging to genotype VI.2.1.1.2.2 and 2 strains belonging to genotype XII.”

- Please mention the references for the viral isolation, HI, Plaque purification. Also please include the reference for the OIE protocol followed for the MDT and ICPI in the reference list.

Response: Thank you for your advice, we have consulted the references you suggest and incorporate them into our manuscript. References for viral isolation, HI, plaque purification, and the MDT and ICPI OIE protocols have been cited.

- Lines 87-89: Why the virulence was between mesogenic and lentogenic while the F protein cleavage site was characteristic of virulent strains? The table shows most of the isolates as velogenic (ICPI >0.7). I think the authors confused lentogenic with

velogenic.

Response: Thanks for your carefully check. The text has been corrected to “Among the 27 strains, according to the OIE criterion “ICPI \geq 0.7 is considered virulent”, 20 strains were virulent strains, and 7 strains had low virulence (Supplementary Table 2).”

- Line 134: "Considering the weaker biosecurity in pigeon industry". Weaker than what?

Response: Thank you for the valuable comment. This sentence was corrected to “Considering the weaker biosecurity of pigeon facilities than that of chicken or swine facilities, this may be one reason for the frequent occurrence of genotype VI infection in domestic pigeon flocks.”

- Please add a conclusion statement at the end.

Response: Thank you for the valuable comment. A conclusion statement was added at the end of the manuscript: “In conclusion, different genotypes, primarily genotype VI NDV, are circulating among multiple species of resident wild birds and may spread back to pigeons. Resident wild birds play an important role in short-distance NDV transmission. More restrictive biosecurity measures and ongoing epidemiological investigations are needed.”

Re: Spectrum00816-24R1 (Analysis of Newcastle disease virus prevalence in wild birds reveals interhost transmission of genotype VI strains)

Dear Dr. Liji Xie:

Your manuscript has been accepted, and I am forwarding it to the ASM production staff for publication. Your paper will first be checked to make sure all elements meet the technical requirements. ASM staff will contact you if anything needs to be revised before copyediting and production can begin. Otherwise, you will be notified when your proofs are ready to be viewed.

Sincerely,
Dharmika Navarathna
Editor
Microbiology Spectrum

Reviewer #1 (Comments for the Author):

The revised manuscript is acceptable. I do not have any more questions

Reviewer #2 (Comments for the Author):

The authors addressed all the comments.